# Automatic Ventriculomegaly Detection in Fetal Brain MRI: A Step-by-Step Deep Learning Model for Novel 2D-3D Linear Measurements

**DOI:** 10.3390/diagnostics13142355

**Published:** 2023-07-13

**Authors:** Farzan Vahedifard, H. Asher Ai, Mark P. Supanich, Kranthi K. Marathu, Xuchu Liu, Mehmet Kocak, Shehbaz M. Ansari, Melih Akyuz, Jubril O. Adepoju, Seth Adler, Sharon Byrd

**Affiliations:** 1Department of Diagnostic Radiology and Nuclear Medicine, Rush University Medical Center, Rush Medical College, Chicago, IL 60612, USA; kranthi_k_marathu@rush.edu (K.K.M.); xuchu_liu@rush.edu (X.L.); mehmet_kocak@rush.edu (M.K.); shehbaz_m_ansari@rush.edu (S.M.A.); melih_akyuz@rush.edu (M.A.); jubril_o_adepoju@rush.edu (J.O.A.); seth_adler@rush.edu (S.A.); sharon_byrd@rush.edu (S.B.); 2Division for Diagnostic Medical Physics, Department of Radiology and Nuclear Medicine, Rush University Medical Center, Rush Medical College, Chicago, IL 60612, USA; mark_supanich@rush.edu

**Keywords:** fetal brain, MRI, artificial intelligence, deep learning, U-Net, ventriculomegaly

## Abstract

In this study, we developed an automated workflow using a deep learning model (DL) to measure the lateral ventricle linearly in fetal brain MRI, which are subsequently classified into normal or ventriculomegaly, defined as a diameter wider than 10 mm at the level of the thalamus and choroid plexus. To accomplish this, we first trained a UNet-based deep learning model to segment the brain of a fetus into seven different tissue categories using a public dataset (FeTA 2022) consisting of fetal T2-weighted images. Then, an automatic workflow was developed to perform lateral ventricle measurement at the level of the thalamus and choroid plexus. The test dataset included 22 cases of normal and abnormal T2-weighted fetal brain MRIs. Measurements performed by our AI model were compared with manual measurements performed by a general radiologist and a neuroradiologist. The AI model correctly classified 95% of fetal brain MRI cases into normal or ventriculomegaly. It could measure the lateral ventricle diameter in 95% of cases with less than a 1.7 mm error. The average difference between measurements was 0.90 mm in AI vs. general radiologists and 0.82 mm in AI vs. neuroradiologists, which are comparable to the difference between the two radiologists, 0.51 mm. In addition, the AI model also enabled the researchers to create 3D-reconstructed images, which better represent real anatomy than 2D images. When a manual measurement is performed, it could also provide both the right and left ventricles in just one cut, instead of two. The measurement difference between the general radiologist and the algorithm (*p* = 0.9827), and between the neuroradiologist and the algorithm (*p* = 0.2378), was not statistically significant. In contrast, the difference between general radiologists vs. neuroradiologists was statistically significant (*p* = 0.0043). To the best of our knowledge, this is the first study that performs 2D linear measurement of ventriculomegaly with a 3D model based on an artificial intelligence approach. The paper presents a step-by-step approach for designing an AI model based on several radiological criteria. Overall, this study showed that AI can automatically calculate the lateral ventricle in fetal brain MRIs and accurately classify them as abnormal or normal.

## 1. Introduction

In fetal central nervous system development, ventriculomegaly (VM) is the most common malformation, with an estimated incidence rate of 7.03 per 10,000 [1]. A fetal brain VM is defined as an atrial width greater than 10 mm perpendicular to the long axis of the lateral ventricle between the frontal horns, cavum septum pellucidum, and glomus of the choroid plexus [1]. Dilatation of the lateral ventricle is usually classified as mild (10–12 mm), moderate (13–15 mm), or severe (>15 mm) depending on the degree of dilatation [2].

Diagnosing and classifying VM prenatally is important because a previous study has shown that VM may indicate other major CNS abnormalities with a sensitivity of 88% [3]. It has been shown that fetal VM may even be associated with morbidity outcomes (especially when outcomes of terminated pregnancies are included) [4]. A smaller VM had a better prognosis than a moderate or severe VM [5]. Once VM has been identified through screening sonography, women are directed to fetal/maternal facilities for further sonographic evaluation, which involves a thorough neuro ultrasound performed by an expert with training experiences in fetal imaging [1]. 

Multiple underlying etiologies complicate the prognosis of fetal VM detection [2]. If other structural CNS abnormalities are detected alongside VM, there is a significant probability of neurologic and/or developmental impairment [3,5].

Abnormality in the cerebral ventricles is one of the most prevalent clinical indications for fetal MRI, mainly to detect additional related abnormalities that may be undiagnosed on prenatal ultrasonography [6]. Even though ultrasound can identify most anomalies, MRI is superior in identifying CNS anomalies due to its higher contrast resolution for brain parenchymal assessment and less sensitivity to fetal positioning, oligohydramnios, maternal obesity, and reverberation artifacts [7].

There are specific requirements that must be met when measuring the size of the lateral ventricle. It is to be measured in the axial plane, in the thalamic section, at the level of the atrium, and at the posterior margin of the glomus of the choroid plexus (Figure 1).

The fetal MRI measures ventricles the same way the US does in the axial plane [8,9]. During pregnancy, the size of the atrium and lateral ventricles stays mostly the same, with ventricular width less than 10 mm considered normal [9].

A 1 to 2 mm difference between US and MRI measurements is considered normal [10]. It is also essential to know that a lateral ventricular asymmetry of 2.4 mm without dilatation is considered normal [11].

Manual measurement of the ventricle size requires specialized training of the physician and can be very time-consuming, given that a fetal MRI exam can contain a large number of 2D series due to repeated acquisitions [4,6]. In addition, because manual measurements are performed on 2D slices, they can potentially contain errors if the 2D slices are not properly oriented, or when the slice with the greatest 2D ventricular width was not correctly identified.

Brain segmentation is the first step in fetal MRI volumetric and morphologic analysis. Automated segmentation could save time and manual labor [7]. Automated brain tissue segmentation in prenatal MRI is difficult due to increased intensity inhomogeneity and spontaneous fetal movements [8]. In recent years, convolutional neural networks (CNNs) have become the most popular medical imaging segmentation method [9].

Segmentation is necessary for fetal brain interpretation. Diagnosis, treatment, and monitoring of neurologic status requires accurate MRI brain tissue segmentation [6].

Over the years, the utilization of deep learning methods, including U-net, has grown significantly in the field of medical image analysis. Deep learning has revolutionized image processing by enabling a range of tasks, including classification, detection, and localization [10]. U-Net, introduced in 2015, demonstrated the benefits of precise segmentation of small objects and its adaptable network structure [11]. Due to the growing demand for high-quality segmentation in medical imaging, the U-Net model has been extensively referenced in academic literature, with over 2500 citations [12].

High precision makes U-Net a popular approach for MR image segmentation [13]. U-net performance is often limited by MRI segmented targets’ varied shapes and information loss from down- and up-sampling. Spatial and channel dimension-based frameworks can solve this problem. Encoding enhances multi-scale features, while decoding recovers localization to a higher resolution layer [14].

In this study, we designed and developed a fully automated workflow that measures the width of the lateral ventricle on fetal MRI images using 3D isotropic data reconstructed from multiple 2D series. We compared the measurement results from the algorithm with those from experienced radiologists.

## 2. Materials and Methods


**Dataset:**


Our institutional IRB approved this retrospective study, and a waiver of informed consent was granted. A total of 22 fetal MRI studies at various gestational ages were retrospectively obtained from fetal MRI exams performed as part of the routine clinical fetal screening at Rush University Medical Center (Chicago, IL, USA) between 2007 and 2020. These studies were selected based on their suitability to be reconstructed into 3D, high-resolution, volumetric fetal brain MR data. Ten of these studies demonstrated normal fetal brain development based on the conclusions of the radiology reports, while the other twelve were diagnosed with ventriculomegaly. Sagittal T2W Half-Fourier Acquisition Single-shot Turbo spin Echo (HASTE) images were exported to a workstation for image quality screening and landmark annotation by an expert pediatric neuroradiologist.


**MRI Protocol:**


All fetal MR exams in this study were performed without sedation on two Siemens MR scanners: a Siemens Sonata 1.5-T MR scanner and a Siemens Espree 1.5-T MR scanner. A 6-channel anterior surface matrix and a posterior spine coil were used to acquire MR signal. All exams included Half-Fourier Acquisition Single-shot Turbo spin Echo (HASTE) acquired in the axial, coronal, and sagittal planes with the following settings: repetition time (TR) = 1400 ms, echo time (TE) = 120 ms, field of view (FOV) = 230 × 230 mm^2^, and slice thickness/gap = 3/0 mm. Patients were breathing normally during the MR exams.


**Automated Workflow for Lateral Ventricular Width Measurement**


The prescribed workflow is summarized in Figure 2 and is elaborated on in subsequent sections. In order to ensure the validity of our automated workflow, we solicited the expert guidance of a highly qualified fetal neuroradiologist to ensure our approach remains in alignment with well-established and clinically endorsed methodologies.

Our model has 7 main steps:

### 2.1. Extracted the Fetal Brain from the Whole Fetal MRI

Images obtained during fetal MR exams are frequently corrupted by motion artifacts resulting from the unpredictable motion of the fetus, as well as that of the pregnant patient. When significant motion artifacts are visible within the acquired 2D MR images, the MR technologist would usually repeat the acquisition of the motion-corrupted series. Therefore, fetal MR exams often contain multiple 2D series acquired within the same anatomic planes. As step 1, the regions representing the fetus’ brain were extracted from all the T2-weighted 2D HASTE series. Figure 3 shows the outputs for fetal brain extraction, in sagittal, coronal, and axial series. 

### 2.2. Sorted Extracted Series According to the Volume (Non-Zero Points)

For each of the three orthogonal (axial, coronal, and sagittal) planes, the segmented 2D series were sorted by the volume of segmented brain tissues, and up to three series with the largest segmented volume were selected as input for the subsequent 3D reconstruction. From these 2D series, image areas representing the brain tissue of the fetus were automatically extracted using MONAIfbs, a deep learning segmentation algorithm proposed by Ranzini et al. that takes advantage of the MONAI framework [15,16]. This method is based on the dynamic UNet (dynUNet) model, which adapts the nnU-Net. Ranzini et al. have shown that when it comes to fetal brain segmentation, the MONAIfbs `method using a single-step approach outperforms the original 2-step approach proposed by Ebner et al. [17].

### 2.3. The 3D Reconstruction from Multiple 2D HASTE Series (Motion Correction and Volumetric Image Reconstruction of 2D Ultra-Fast MRI)

After the brain tissues were segmented from the 2D brain HASTE series, we applied the NiftyMIC algorithm, an automated framework developed by Ebner et al., to reconstruct the three low-resolution, 2D image stacks into a high-resolution, 3D isotropic representation of the fetal brain at 0.5 mm voxel size, with necessary angular correction so that the 3D brain volume accurately aligns with the standard anatomic planes.

The 3D reconstruction from multiple 2D HASTE series: For each exam, five 2D brain input stacks were selected from the sorted list (Step 2): three axial, one coronal, and one sagittal.

We have selected three axial series because the axial plane is also used to measure ventricles in clinical practice. Figure 4 shows the output of the reconstruction. 

### 2.4. Segmentation of Fetal Brain to Seven Tissues

In this step of our study, we implemented the AI model adopted by the winning team of the FeTA 2021 challenge (Team name NVAUTO) to perform automatic fetal brain tissue segmentation. The AI model is based on the MONAI ResSegNet with OCR modules. The reconstructed 3D brain volume is segmented into seven tissue categories (External Cerebrospinal Fluid, Grey Matter, White Matter, Ventricles, Cerebellum, Deep Grey Matter, and Brainstem/Spinal Cord). A summary of the original model (NVAUTO) and the model adapted to our implementation (RUSH) is described in Table 1. After its assembly, the deep learning network was trained with the collection of annotated fetal MR data from the FeTA 2022 challenge.

Once we achieved satisfactory segmentation results with the annotated FeTA 2022 dataset, we deployed the trained MONAI ResSegNet to the 3D MR data reconstructed from our institution’s 22 fetal MR exams. We manually inspected the results of these segmentations, and found the segmentations for all 22 exams to be generally satisfactory. 

### 2.5. Defining the Maximum of “Deep Gray Matter”, as a Clue for Finding the Best Slice for Measuring Ventricle

Following the automatic segmentation, the axial slice with the largest segmented area of Deep Gray Matter from the high-resolution 3D MR data is selected as the reference slice from which the width of ventricles will be measured (Figure 5) because the thalamus is included in the DGM category in the brain segmentation of the model. This is consistent with the protocol of manual measurement performed by physicians (Appendix A).

### 2.6. Automatic Linear Measurement of the Lateral Ventricle

When manual measurement is performed, the lateral ventricular width is measured at the level of the atrium, and at the posterior margin of the glomus of the choroid plexus on the axial plane through the thalamus. Our automated workflow is designed to measure the ventricular width in a manner consistent with clinical practice.

#### 2.6.1. Extraction of a Single Ventricle (Left Ventricle or Right Ventricle)

In this step, we focus on extracting a single ventricle from the reference slice. To accomplish this, we employ a connected-component labeling algorithm on the segmented “ventricle.” The output is divided into three distinct parts: (1) Left ventricle, (2) Right ventricle, and (3) septum pellucidum. Our workflow proceeds by addressing each ventricle separately, repeating steps 2.6.2 to 2.6.3 for both the Left and Right ventricles.

#### 2.6.2. Removal of Segmentation Errors and Choroid Plexus Elimination (Using Binarization Function)

The segmented lateral ventricles exhibit small margins along their boundaries (Figure 6a), which render them unsuitable for accurate width measurement. To rectify this, we apply a thresholding process to determine the average pixel values within the segmented ventricles. Subsequently, all pixels with values below the threshold are excluded from the ventricle segment, effectively eliminating segmentation errors and removing the choroid plexus (Figure 6b represents our mask, while Figure 6c illustrates the resulting output).
$$I_{binary}\left(i,j\right)=\left\{\begin{array}{c} 1,\;if\;I\left(i,j\right)\geq mean\\ 0,\;otherwise \end{array}\right.$$

#### 2.6.3. Rotating the Ventricle for Linear Measurement

In this step, we perform a rotation of the image to facilitate the calculation of ventricle width. Using OpenCV’s prebuilt algorithms, we identify a minimal rectangular bounding box that encompasses the refined ventricular segment (Figure 6a–c). By examining this bounding box, we can easily determine the major and minor axes of the ventricles, allowing us to reorient the ventricular segments angularly. This reorientation aligns their axes orthogonally with those of the 2D image. Subsequently, the ventricular width is obtained by calculating the length of the *Y*-axis, representing the vertical direction (Figure 7).

### 2.7. Manual Measurement of Lateral Ventricular Width

To evaluate the accuracy of the ventricle width measured through our automated workflow, we enlisted the assistance of two experienced neuroradiologists to manually calculate the width of the lateral ventricles in 22 fetal MRI exams (see Appendix A). It is important to note that, unlike our automated workflow, the manual measurements were not conducted on the reconstructed 3D image but rather on a single slice from the 2D series the radiologists chose as most appropriate. The results of the manual measurements, recorded in millimeters for each of the 22 fetal MRI exams, were then compared to the measurements obtained through our automated workflow.

Figure 8 provides an illustrative example showcasing the comparison between AI and manual measurements.

## 3. Results

### 3.1. Normal vs. Abnormal Classification

Out of the total 22 cases, 10 had ventricles that measured less than 10 mm on both the right and left sides and were therefore categorized as normal. The remaining 12 cases showed ventricles that measured more than 10 mm in both sides, which were classified as abnormal. Based on this categorization, our automated workflow was able to identify ventriculomegaly in 21 out of the 22 cases, resulting in an accurate classification rate of 95%, as shown in Table 1

### 3.2. AI Measurement in Normal Cases

Out of the ten exams, and 20 ventricles that were classified as normal, AI was able to accurately measure the ventricle size with an average error of less than 1.7 mm, resulting in a 100% accuracy in the classification of normal cases (comparing both a general radiologist and neuroradiologist). Table 2 shows AI vs. manual measurements of the lateral ventricle in normal cases.

We set 1.7 mm as a threshold of error because it was the maximum error of our accepted measurements according to the expert pediatric neuroradiologist’s opinion (Dr. Sharon Byrd).

### 3.3. Measurement in Abnormal Cases

Among the 12 cases that were classified as abnormal, a total of 24 ventricles (both left and right) were identified. AI was able to measure the size of 21 out of the 24 abnormal ventricles with an error rate of less than 1.7 mm, resulting in an accuracy rate of 87.5% in measuring ventricle size in abnormal cases. Table 3 presents a detailed list of the abnormal cases, while Figure 9 provides an example of how the AI measurements compared to manual measurements in abnormal cases.

An example of AI vs. manual measurement in an abnormal case is shown in Figure 9.

### 3.4. Measurement of R2 Score in Right Ventricle

We calculated the R2 score to evaluate the correlation between the predictions made by the general radiologist, neuroradiologist, and AI. For the right ventricle, the R2 score was 0.94 for the comparison between the general radiologist and AI, and 0.96 for the comparison between the neuroradiologist and AI. Figure 10a presents a box plot that compares the AI, general radiologist, and neuroradiologist predictions for the right ventricle, while Appendix A provides a scatter for these comparisons.

### 3.5. Measurement of R2 Score in Left Ventricle

For the left ventricle, the R2 score was 0.93 for the general radiologist vs. AI; the R2 score was 0.92 for the neuroradiologist vs. AI. The box plot in Figure 10b compares the AI with the general radiologist vs. the neuroradiologist for the left ventricle (Appendix A is the scatter for these comparisons).

Table 4 shows the error difference between manual measurements (Radiologist and neuroradiologist) compared with AI measurements.

### 3.6. Comparing t-Test for Different Measurements

The paired t-test was applied to compare the measurement of the 44 ventricles in three different scenarios:General radiologist vs. NeuroradiologistGeneral radiologist vs. AINeuroradiologist vs. AI

The results showed that there was no statistical difference in scenario 2 (between general radiologist vs. AI) (*p* = 0.9827), and scenario 3 (neuroradiologist vs. AI) (*p* = 0.2378). However, the difference in scenario one (general radiologist vs. neuroradiologist) was statistically significant (*p* = 0.0043). This highlighted the performance of the AI measurement of the ventricle.

## 4. Discussion

The primary objective of this research project was to develop an automated method to measure the size of lateral ventricles in fetal brain MRIs using deep learning (DL) algorithms. To accomplish this, we adapted a U-net based segmentation model and trained it using a publicly available labeled dataset (FeTA 2022) that consisted of 80 normal and abnormal T2-weighted fetal MR images, in which the fetal brains were segmented into seven different brain tissues. We then tested the trained algorithm using a dataset comprising 22 T2-weighted fetal brain MRI exams from our institution. The exams used to test our algorithm were also classified into normal and abnormal cases based on manual measurements performed by an expert neuroradiologist.

Our AI model accurately classified fetal brain MRI cases as normal or ventriculomegaly with an accuracy rate of 95%. Furthermore, the AI model was able to measure the lateral ventricle diameter in 100% of normal ventricles and 87.5% of abnormal ventricles with an error rate of less than 1.7 mm.

The difference between measurements was 0.90 mm in AI vs. General radiologists. This difference was 0.84 mm in AI vs. neuroradiologists. Therefore, AI measurements were closer than neuroradiologists’ measurements.

Additionally, we were able to reconstruct 3D images, which are closer to real anatomy. It could also provide both right and left ventricles in one cut (compared to two cuts in manual measurement). The following sections describe our AI model’s novelty, advantages, and limitations [18,19].


**Novelty of our AI model for ventriculomegaly:**


1. For the first time, we developed an AI model that can perform the “Linear” measurement of the lateral ventricle. All similar papers (ref) used a 3D model, which is not useful for clinical use because 3D volumetry does not correlate directly with ventriculomegaly (which is measured linear 2D in clinical practice)

2. Extracting 2D linear measurements from 3D constructed images requires additional AI design, which was done by specifying several rules (one for the thalamus, another for the Choroid plexus, and one for the axis of rotation and axis of ventriculi.)

3. We achieved good agreement between our output and the measurements made by neuroradiologists by considering the Thalamus and Choroid plexus, which are radiological clues. It was validated by an expert fetal neuroradiologist, as well as a neuroradiologist fellowship and general radiologist physician; all measurements of AI for the lateral ventricles were acceptable and accurate.

4. In this project, we performed two main tasks: Task 1: Brain Extraction, Reconstruction, and Segmentation; Task 2: Automated and linear measurements in the segmented ventricles (Using different Python libraries like Using the OpenCV library).

As each of these steps had technical and radiological challenges, we described our solutions in detail step-by-step in the paper.

For example, we used the “binarization function” to detect the choroid plexus in task 2. Furthermore, we improved the accuracy of ventricle segmentation by defining rules to remove black signals around the ventricles (the left picture has some extra black signals around the ventricle, which was caused by the segmentation problem; we solved this problem by calculating the “average gray value of the image as the threshold).

Along with detecting the choroid plexus (which always has a black signal), this solution also automatically removes additional black signals around the ventricle as well. A further novelty of our project was using the “Minimum Area Rectangle “to determine the appropriate angle for linear measurements in the lateral ventricle [7,9,14,20].


**Similar Studies:**


Fetal MRI combined with fetal ultrasound increases confidence in the early detection of perinatal disorders of development [21]. Manual measurements have several disadvantages, including clinician training requirements, time commitment, and inter- and intra-observer variability [22,23]. AI and deep learning have several capabilities in the automatization of fetal brain MRI tasks, which we review in a separate paper [8].

Different applications for AI in fetal brain MRI include preprocessing images to assess quality, detect landmarks, correct fetal motion, and predict motion in real-time [24,25,26]. AI is also used for post-processing tasks such as autonomously segmenting normal fetal brains and performing super-resolution reconstruction [17,27]. Fetal imaging can be reconstructed through a fully automatic framework consisting of multiple stages [17]. AI models are employed for gestational age prediction using brain MRI [28] and classifying brain pathology [29]. AI models have been developed for functional fetal brain MRI [30].

The previous papers discussed various segmentation strategies and models for automated fetal brain segmentation in MRI. Makropoulos et al. provided an overview of different segmentation approaches, including unsupervised, parametric, classification, atlas fusion, and deformable models [31]. CNN architectures, such as U-Net, have gained popularity for medical image segmentation, as demonstrated by studies conducted on adult MRIs (Mohseni Salehi et al. [32], and Rampun et al. [33]). Khalili et al. proposed a CNN approach using augmented images with synthetically induced intensity inhomogeneity for segmentation. Tourbier et al. introduced a pipeline for localizing, segmenting, and reconstructing intracranial volumes (ICV) in fetal MR data [34]. Similarly, Link et al. developed a semi-automatic fetal brain segmentation approach using MRI data and a volumetric growth chart [35].

Also, linear measurements have been done in different designs in previous papers. Avisdris et al. developed a deep-learning method to automatically compute linear measurements in fetal brain MRI volumes using landmark detection [20]. This fully automated method computed three key fetal brain MRI parameters: Cerebral Biparietal Diameter (CBD), Bone Biparietal Diameter (BBD), and Trans Cerebellum Diameter (TCD). Their model has a 95% confidence interval agreement of 3.70 mm for CBD, 2.20 mm for BBD, and 2.40 mm for TCD compared to a fetal radiologist. In another project for automatic measurement of pons and vermis in fetal brain MRI, Vahedifard et al. developed the Unet model. In 85% of cases, confidence levels were within 90%, with a mean error of 2.22 mm [36,37].

While the previous papers primarily focused on general fetal brain segmentation, we specifically addressed ventriculomegaly detection using a step-by-step deep learning model. By applying AI to perform 2D linear measurements of ventricles, our study demonstrated the potential of AI in automatically assessing ventriculomegaly with high accuracy. We showcased the design process of our AI model based on radiological criteria and emphasized the clinical relevance of AI-enabled ventricle calculation in fetal brain MRI. Our study highlighted the feasibility and accuracy of AI-based ventricle measurement in fetal brain MRIs, offering a novel approach in this domain.

Overall, the previous papers contribute to the broader field of fetal brain segmentation, while our paper provides a focused investigation on ventriculomegaly detection using an AI-based approach. The combination of automated segmentation and accurate ventricle measurement demonstrates the potential of AI to improve diagnostic capabilities and streamline the evaluation of fetal brain abnormalities in clinical practice.

## 5. Limitation

In this model, 3D reconstruction is used to achieve the highest level of prediction accuracy. As a result, this AI model cannot be used if the original images do not have enough quality for 3D reconstruction.

The time required for 3D reconstruction was another limitation: In the first design, 9 image series were used, according to the description of the original tools. Next, we tried to reduce the time by selecting 3–5 image series in our reconstruction tools. By doing so, we reduced reconstruction time from 40–60 min to 20–30 min.

It should be noted that reconstruction was only one of several steps that were performed in this AI project. The step-by-step linear measurements were the main objective of the research. Additionally, a faster computer will speed up reconstruction. To reconstruct the image, we used an HP Z4 G4 Workstation IDS Base Model with an NVIDIA GeForce RTX 2080 Ti Graphics Card.

## 6. Conclusions

We developed a DL model for automatically measuring the lateral ventricle in fetal brain MRI and classifying them as normal or abnormal. Based on our Fetal Brain MRIs results, we could classify the cases as being normal or showing ventriculomegaly with 95% accuracy.

Also, AI could measure the lateral ventricle diameter in 95% of cases with less than a 1.7 mm error. The difference between measurements was 0.90 mm in AI vs. General radiologists. This difference was 0.84 mm in AI vs. neuroradiologists. AI measurements were closer to neuroradiologists’ measurements, rather than general radiologists.

This paper presented a step-by-step approach to designing an AI model based on several radiological criteria. By following several steps, AI-prediction results are very similar to real-life scenarios.

## Figures and Tables

**Figure 1 diagnostics-13-02355-f001:**
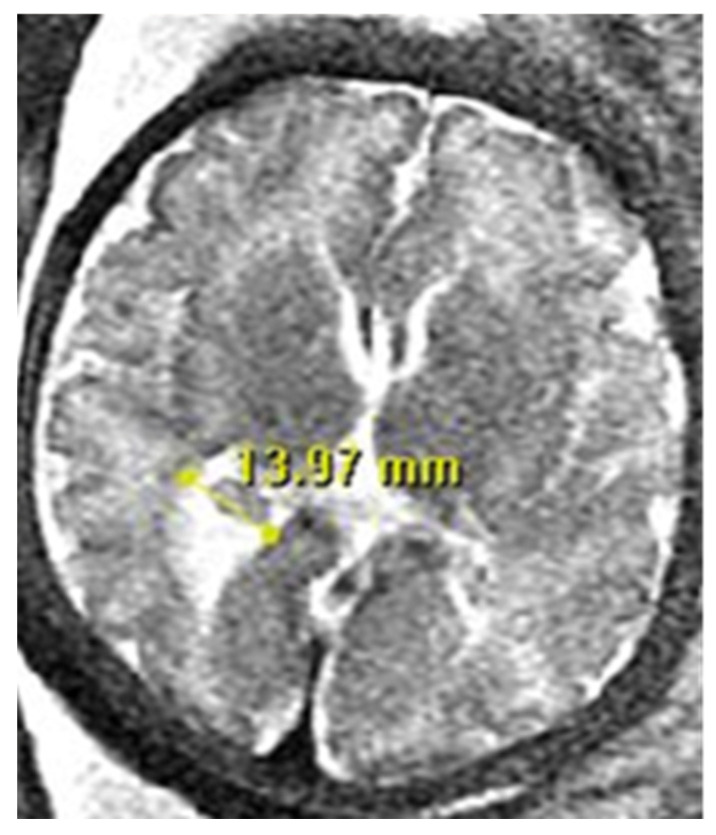
Neuroradiologists’ criteria for measuring the lateral ventricle in Fetal Brain MRI: Lateral ventricular measurement is measured at the level of the atrium at the posterior margin of the glomus of the choroid plexus, on the axial plane through the thalamus.

**Figure 2 diagnostics-13-02355-f002:**
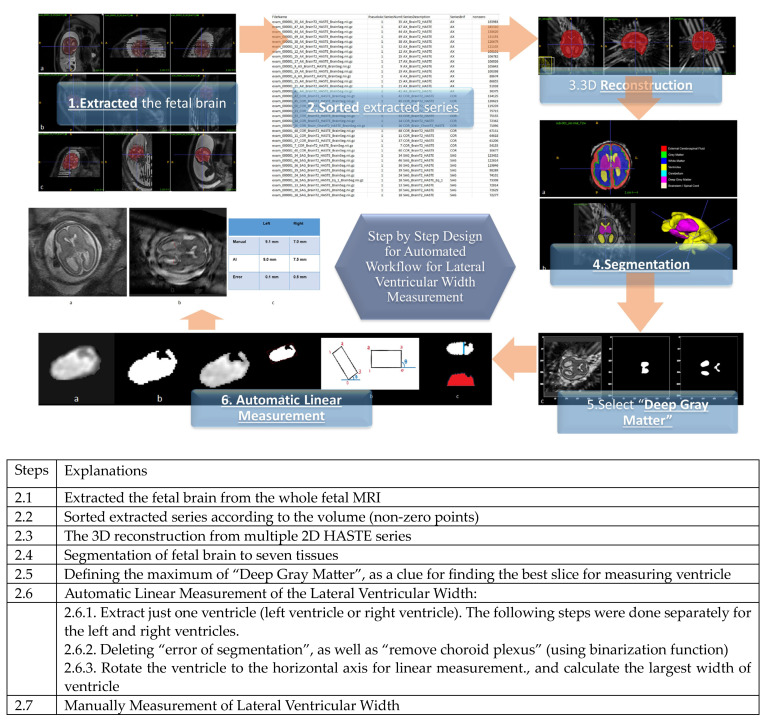
Summary of the workflow developed in our study for the automatic measurement of the size of the lateral ventricle.

**Figure 3 diagnostics-13-02355-f003:**
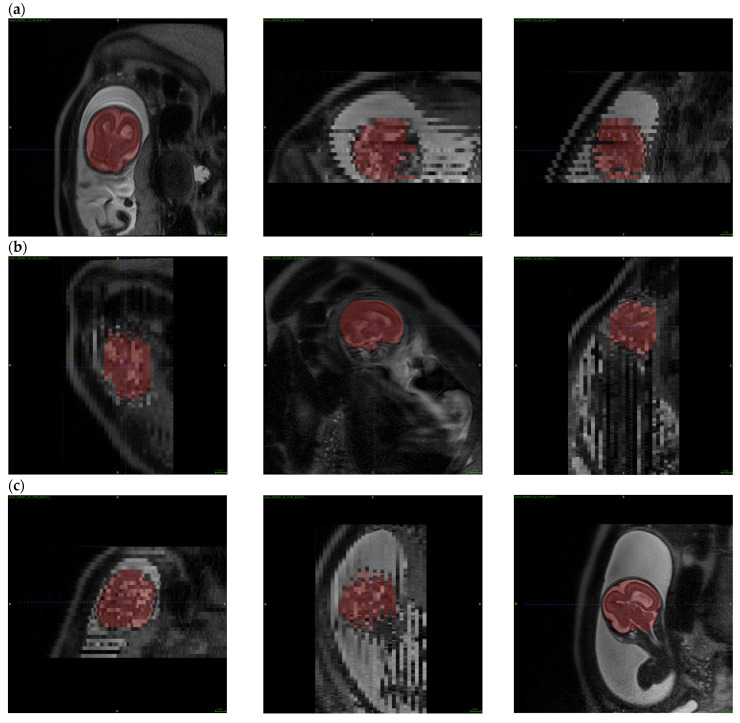
An example of fetal brain extraction and segmentation in all three anatomic orientations. (**a**) is axial, (**b**) is sagittal, and (**c**) is coronal for fetal brain segmentation.

**Figure 4 diagnostics-13-02355-f004:**
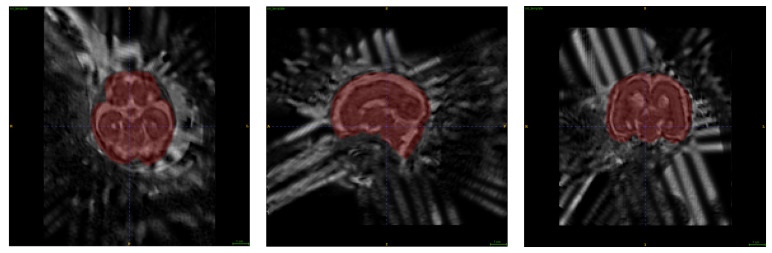
3D Reconstruction: A Qualitative Comparison between Low-Resolution Input Data and High-Resolution Volumetric Reconstructions.

**Figure 5 diagnostics-13-02355-f005:**
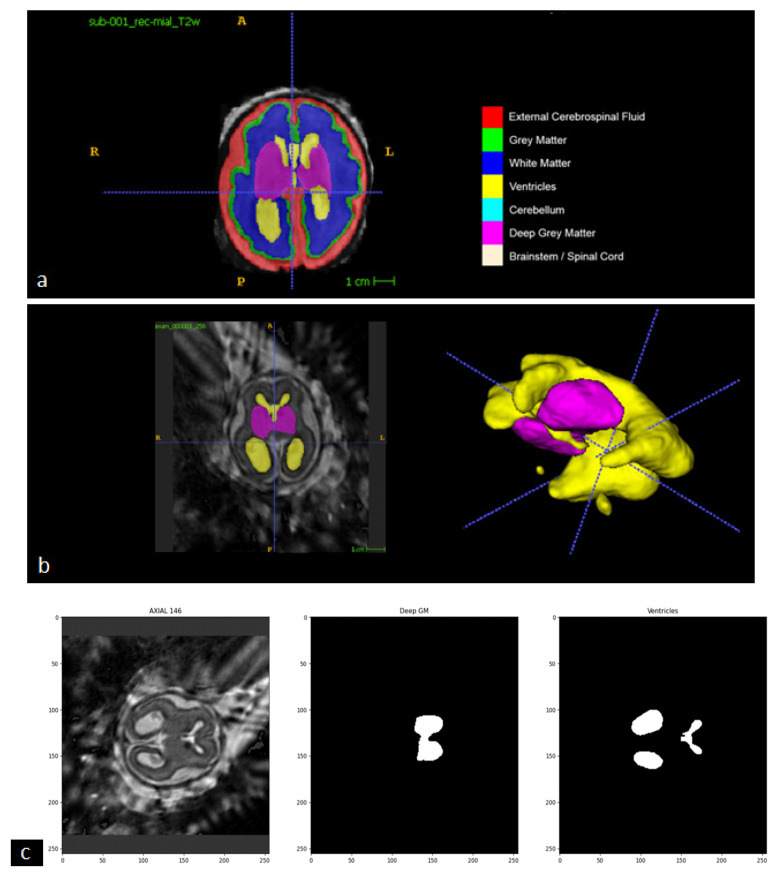
Key Steps in Automatic Linear Measurement of the Lateral Ventricle. (**a**) General Picture of the Online Dataset: The online dataset is divided into seven different tissues for analysis. (**b**) Utilizing Deep Gray Matter for Ventricle Identification: The Deep Gray Matter (purple segment) is used as a reference to locate and identify the ventricle (yellow segment) in the 3D version. (**c**) Defining Deep Gray Matter as a Clue for Ventricle Measurement: The Deep Gray Matter, specifically the Thalamus, serves as a crucial clue in selecting the optimal cut for accurate ventricle measurement. The image includes a 3D reconstructed image slice (left side), the Deep Gray Matter (middle), and the extracted ventricle (right side).

**Figure 6 diagnostics-13-02355-f006:**
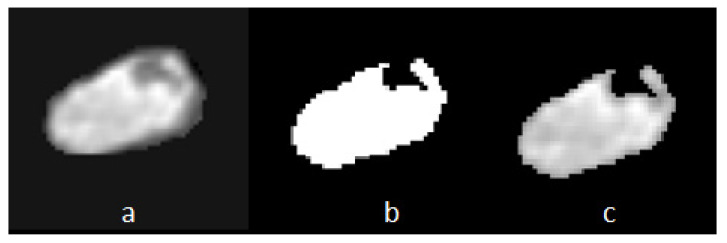
Binarization Function for Segmentation Error Removal and Choroid Plexus Elimination. (**a**) Ventricle with Segmentation Error: The ventricle segmentation exhibits small margins along its boundaries, hindering accurate width measurement. (**b**) Binarization Mask: A mask is obtained through the application of a binarization function, highlighting the desired ventricle region. (**c**) Modified Ventricle Segmentation: The modified ventricle segmentation (c = a ∩ b) represents the intersection between the original ventricle (**a**) and the binarization mask (**b**), effectively eliminating segmentation errors and excluding the choroid plexus.

**Figure 7 diagnostics-13-02355-f007:**
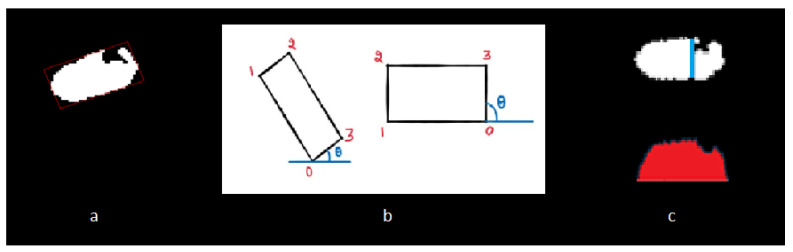
Rotation of the Segmented Ventricle for Linear Measurement. (**a**) Enclosing Square: The square that encompasses the ventricle is identified for accurate measurement. (**b**) Ventricle Axis Definition: The ventricle axis is established to calculate the angle required for drawing the line. The left side represents the axis before binarization, while the right side shows the axis after binarization. The numbers can help to locate the corners of box, for proper rotations. (**c**) Line Adjustment to Choroid Plexus: The line is moved from the largest diameter towards the choroid plexus. The left side shows the original line position, while the right side depicts the line adjusted to touch the choroid plexus.

**Figure 8 diagnostics-13-02355-f008:**
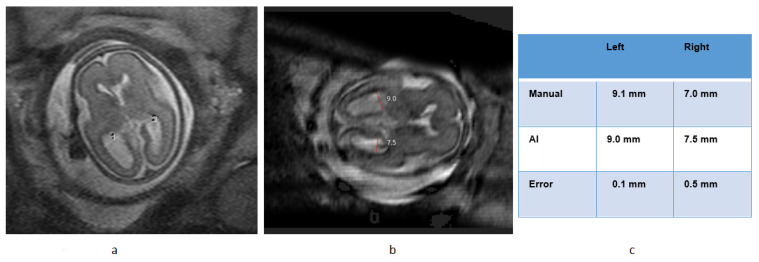
Example of ventricle measurement in Normal Cases using AI. (**a**) Manual measurement, (**b**) AI- measurement, (**c**) Comparison table.

**Figure 9 diagnostics-13-02355-f009:**
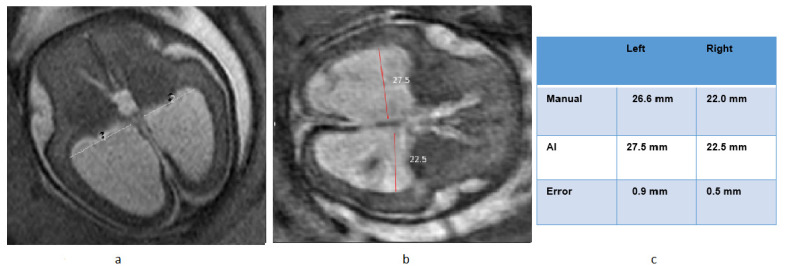
Sample of ventricle measurement in Abnormal Cases using AI. (**a**) Manual measurement, (**b**) AI- measurement, (**c**) Comparison table.

**Figure 10 diagnostics-13-02355-f010:**
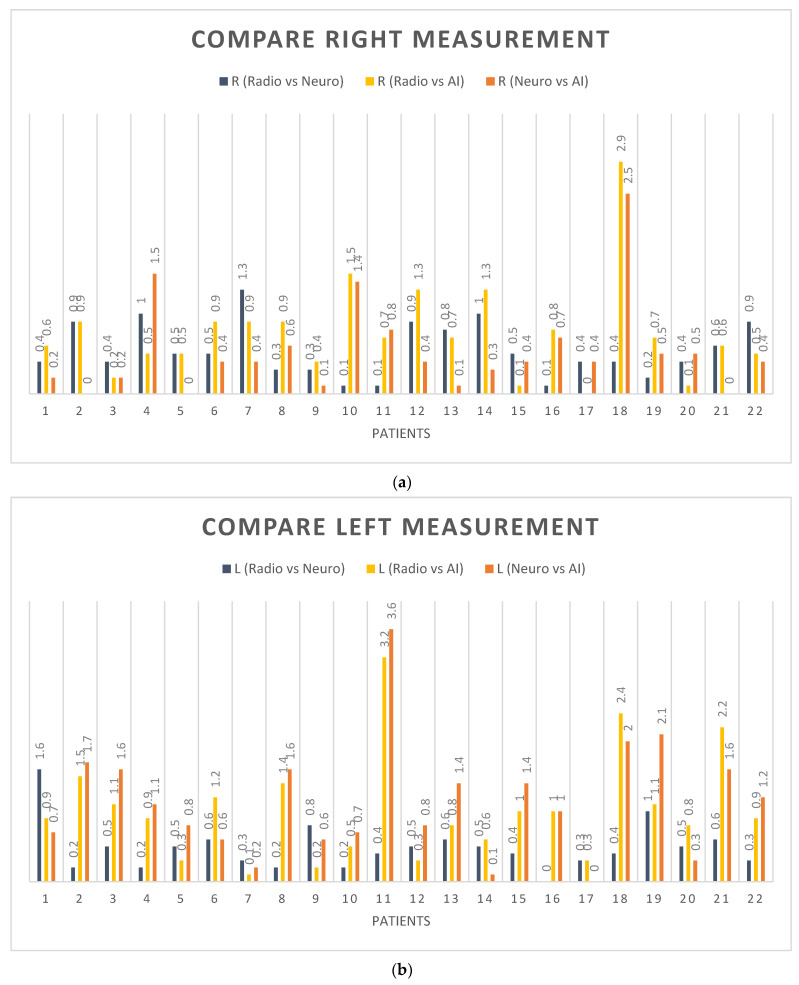
(**a**) Boxplot for comparing the Fetal Brain MRI Right ventricle’s measurements by the AI, radiologist, and neuroradiologist (Right R2 (Radiologist vs. AI) = 0.948159231; Right R2 (Neuroradiologist vs. AI) = 0.964945795)). (**b**) Boxplot for comparing the Fetal Brain MRI Left ventricle’s measurements by the AI, radiologist, and neuroradiologist.

**Table 1 diagnostics-13-02355-t001:** Information from the original AI model (NVAUTO), which we modified and used in this paper.

Team Name	Network	Loss Function	2D/3D	Patch Size	Post-Processing	Convolution Kernel Size	Optimizer
NVAUTO	MONAI[SegResNet], OCR modules	Dice	3D	224 × 224 × 144	Ensemble learning	3 × 3 × 3	AdamW
RUSH	MONAI[SegResNet]	Dice	3D	224 × 224 × 144	None	3 × 3 × 3	Adam
**Team Name**	**Initialization**	**Learning Rate**	**Cross-Validation**	**Epochs**	**GPU Used**	**# of Layers**	**# of Trainable Parameters**
NVAUTO	Random	0.0002, decrease to 0 at final epoch with cosine annealing scheduler	5-fold	300	4 × vidia V100 32G	5desc/5asc	75,819,624
RUSH	RandSpatialCrop RandFlip RandScaleIntensity RandShiftIntensity	0.0001, weight_decay = 0.00001, CosineAnnealingLR	5-fold	300	Nvidia GeForce RTX 2080 Ti 12G	5desc/5asc	4,700,999

**Table 2 diagnostics-13-02355-t002:** Result of AI vs. manual measurements of the lateral ventricle in normal cases.

Patient’s Number	Right Ventricle (Manual-General Radiologist)	Left Ventricle (Manual-Radiologist)	Right Ventricle (Manual-Neuroradiologist)	Left Ventricle (Manual-Neuroradiologist)	Right Ventricle (AI-Predicted)	Left Ventricle (AI-Predicted)
1 (normal)	5.1	7.9	4.7	6.3	4.5	7.0
2 (normal)	5.6	8.5	6.5	8.7	6.5	7.0
3 (normal)	6.7	5.4	6.3	4.9	6.5	6.5
4 (normal)	7.0	9.1	8.0	8.9	6.5	10.0
5 (normal)	7.0	8.7	7.5	8.2	7.5	9.0
6 (normal)	7.4	5.3	6.9	5.9	6.5	6.5
7 (normal)	8.1	5.9	9.4	6.2	9.0	6.0
8 (normal)	8.6	8.4	8.9	8.6	9.5	7.0
9 (normal)	8.6	8.8	8.9	9.6	9.0	9.0
10 (normal)	9.0	8.5	8.9	8.7	7.5	8.0
Mean	7.31	7.65	7.6	7.6	7.3	7.55
SD	1.29	1.49	1.50	1.60	1.53	1.40

**Table 3 diagnostics-13-02355-t003:** Result of AI vs. manual measurements of the lateral ventricle in abnormal cases.

Patient’s Number	Right Ventricle (Manual-General Radiologist)	Left Ventricle (Manual-Radiologist)	Right Ventricle (Manual-Neuroradiologist)	Left Ventricle (Manual-Neuroradiologist)	Right Ventricle (AI-Predicted)	Left Ventricle (AI-Predicted)
11 (abnormal)	7.2	12.7	7.3	13.1	6.5	9.5
12 (abnormal)	10.2	5.3	11.1	5.8	11.5	5.0
13 (abnormal)	10.3	9.8	11.1	10.4	11.0	9.0
14 (abnormal)	10.3	5.6	9.3	5.1	9.0	5.0
15 (abnormal)	10.6	12.0	10.1	12.4	10.5	11.0
16 (abnormal)	11.3	10.5	11.2	10.5	10.5	9.5
17 (abnormal)	12.0	13.8	12.4	13.5	12.0	13.5
18 (abnormal)	12.1	12.1	12.5	12.5	15.0	14.5
19 (abnormal)	12.3	14.1	12.5	15.1	13.0	13.0
20 (abnormal)	14.1	12.8	14.5	12.3	14.0	12.0
21 (abnormal)	16.9	15.3	17.5	15.9	17.5	17.5
22 (abnormal)	22.0	26.6	22.9	26.3	22.5	27.5
Mean	12.18	12.75	12.64	12.81	11.22	10.79
SD	4.19	5.10	5.08	5.31	3.31	4.22

**Table 4 diagnostics-13-02355-t004:** The difference between manual measurements (General radiologist and neuroradiologist) compared with AI measurements.

Patient’s Number	Right Ventricle (General Radiologist vs. Neuroradiologist)	Left Ventricle (General Radiologist vs. Neuroradiologist)	Right Ventricle (General Radiologist vs. AI)	Left Ventricle (General Radiologist vs. AI)	Right Ventricle (Neuroradiologist vs. AI)	Left Ventricle (Neuroradiologist vs. AI)
1 (normal)	0.4	1.6	0.6	0.9	0.2	0.7
2 (normal)	0.9	0.2	0.9	1.5	0.0	1.7
3 (normal)	0.4	0.5	0.2	1.1	0.2	1.6
4 (normal)	1.0	0.2	0.5	0.9	1.5	1.1
5 (normal)	0.5	0.5	0.5	0.3	0.0	0.8
6 (normal)	0.5	0.6	0.9	1.2	0.4	0.6
7 (normal)	1.3	0.3	0.9	0.1	0.4	0.2
8 (normal)	0.3	0.2	0.9	1.4	0.6	1.6
9 (normal)	0.3	0.8	0.4	0.2	0.1	0.6
10 (normal)	0.1	0.2	1.5	0.5	1.4	0.7
11 (abnormal)	0.1	0.4	0.7	3.2	0.8	3.6
12 (abnormal)	0.9	0.5	1.3	0.3	0.4	0.8
13 (abnormal)	0.8	0.6	0.7	0.8	0.1	1.4
14 (abnormal)	1.0	0.5	1.3	0.6	0.3	0.1
15 (abnormal)	0.5	0.4	0.1	1.0	0.4	1.4
16 (abnormal)	0.1	0.0	0.8	1.0	0.7	1.0
17 (abnormal)	0.4	0.3	0.0	0.3	0.4	0.0
18 (abnormal)	0.4	0.4	2.9	2.4	2.5	2.0
19 (abnormal)	0.2	1.0	0.7	1.1	0.5	2.1
20 (abnormal)	0.4	0.5	0.1	0.8	0.5	0.3
21 (abnormal)	0.6	0.6	0.6	2.2	0.0	1.6
22 (abnormal)	0.9	0.3	0.5	0.9	0.4	1.2

Mean of errors	0.55	0.48	0.77	1.03	0.54	1.14
Standard Deviation	0.34	0.33	0.62	0.76	0.59	0.82
Mean of errors (right and left)	0.51 mean error for Right and left (General Radiologist vs. Neuroradiologist)		0.90 mean error for Right and left (General Radiologist vs. AI)		0.84 mean error for Right and left (Neuroradiologist vs. AI)	

## Data Availability

Source code used in this project, which are supporting reported results can be found in the GitHub of Rush University Medical Center: GitHub—marksupanich/RUSHRadiologyResearch: Repository for RUSH Radiology Research.

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
