# Peer review of "Automatic Ventriculomegaly Detection in Fetal Brain MRI: A Step-by-Step Deep Learning Model for Novel 2D-3D Linear Measurements"

_diagnostics, 2023, doi:10.3390/diagnostics13142355_

Round 1

Reviewer 1 Report

The paper is written well, and we developed an automated workflow using a deep learning model (DL) to measure the lateral ventricle linearly in fetal brain MRI, which is subsequently classified into normal or ventriculomegaly, defined as a diameter wider than 10 mm at the level of the thalamus and choroid plexus.

I am wondering why they used segmentation models from other teams. Why they did not propose their own segmentation models?

Did authors try nnUNet for segmentation and compare it with their own methods?

Is there any specific reason to validate their method of feta dataset?

The authors should open their code to reproduce the results. The references in this paper are too short. It should be added more.

Reviewer 2 Report

Esteemed Author Team,

Manuscript ID: diagnostics -2480993 is an interesting, well-written, original research article describing a step-by-step approach to design an AI model based on several radiological criteria in normal and abnormal fetal MRI studies.

Few isolated suggestions and remarques are listed below:

1.Table 2 in line 154 should be most probably table 1. Same error in line 204 - again I assume it should be (Table 1), and not (Table 2). In summary, there are two different tables, numbered Table 3, the first one preceding Table 2. The second table 3 should be, in fact, table 4.  Please, correct the numbering of the tables!

2. MONAI abbreviation may appear first at line 153 instead of line 161. "3D reconstruction" in line 179 is underlined, why?

3. The statement in page 6, lines 182-187:"Images obtained during routine fetal MR exams are frequently corrupted by motion artifacts resulting from the unpredictable motion of the fetus, as well as that from the pregnant patient. When significant motion artifacts are visible within the acquired 2D MR images, the MR technologist would usually repeat the acquisition of the motion-corrupted series. Therefore, fetal MR exams often contain multiple 2D series acquired within the same anatomic planes" is repeated from the beginning of the paragraph describing the Brain Tissue Segmentation with MONAI (line144-148), is it really necessary or can be avoided?

4. The sentence in line 211 is underlined, unfinished and inconsistent?

6. Subtle symbols errors -lines 217/218 - missing and double "." Same errors are found in the Discussion

7. Page 11, in line 313 replace Figure 8 by Figure 9.

8. First sentence in the Discussion, correct to "objective".

9. You are using a 6-channel surface body coil to acquire fetal MR studies. Do you consider to obtain better resolution, therefore better quality of segmentation by using multi-channel coils?

General comments: English language is fine. The methods are adequately described and  the results are clearly presented. Please, check again the instructions of the journal for the references and correct them. The number of the references is limited.

Overall, I find the data valuable and only minor corrections are needed prior to proceed to publication. The Introduction could be slightly shorter. The Discussion repeat some  paragraphs from the Results. Adding few more references and/or comparison to similar studies would greatly benefit to improve the Discussion part and the overall impression of the manuscript.

It will be nice if those subtle changes are clearly visible as Track Changes On. Thank you!
